# Inferring Smooth Control: Monte Carlo Posterior Policy Iteration with Gaussian Processes

**Joe Watson**[1]      **Jan Peters**[1234]

[1]Department of Computer Science, Technical University of Darmstadt
[2]Centre for Cognitive Science, Technical University of Darmstadt
[3]German Research Center for AI      [4]Hessian.AI
{joe,jan}@robot-learning.de

**Abstract:** Monte Carlo methods have become increasingly relevant for control of non-differentiable systems, approximate dynamics models and learning from data. These methods scale to high-dimensional spaces and are effective at the non-convex optimizations often seen in robot learning. We look at sample-based methods from the perspective of inference-based control, specifically posterior policy iteration. From this perspective, we highlight how Gaussian noise priors produce rough control actions that are unsuitable for physical robot deployment. Considering smoother Gaussian process priors, as used in episodic reinforcement learning and motion planning, we demonstrate how smoother model predictive control can be achieved using online sequential inference. This inference is realized through an efficient factorization of the action distribution and a novel means of optimizing the likelihood temperature to improve importance sampling accuracy. We evaluate this approach on several high-dimensional robot control tasks, matching the sample efficiency of prior heuristic methods while also ensuring smoothness. Simulation results can be seen at `monte-carlo-ppi.github.io`.

**Keywords:** approximate inference, policy search, model predictive control

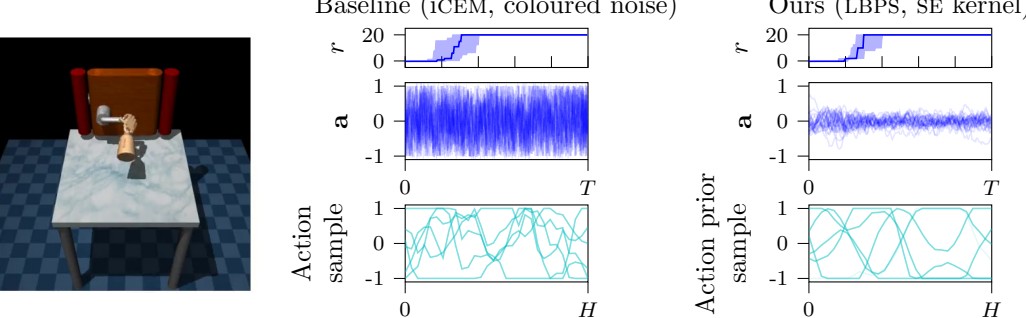

Figure 1: High-dimensional, contact-rich tasks such as manipulation (left) can be performed effectively using sample-based model predictive control. While prior work uses correlated actuator noise to improve sample-efficiency and exploration, these methods do not preserve the smoothness in the downstream actuation $\boldsymbol{a}$, resulting in aggressive control (center). We use smooth Gaussian process priors to infer posterior actions (right), which preserves smoothness while maintaining performance and sample efficiency, as both are using only 32 samples. Rewards $r$ show quartiles over 25 seeds.

## 1  Introduction

Learning robot control requires optimization to be performed on sampled transitions of the environment [1]. Monte Carlo methods [2] provide a principled means to approach such algorithms, bridging black-box optimization and approximate inference techniques. These methods have been adopted extensively by the community for their impressive simulated [3, 4, 5, 6] and real-world [7, 8, 9, 10, 11, 12, 13, 14, 15] robot learning results. Their appeal includes requiring only function evaluations of the dynamics and objective, so can be applied to complex environments with minimal overhead (Figure 1). Moreover, their stochastic nature also avoids issues with local minima

6th Conference on Robot Learning (CoRL 2022), Auckland, New Zealand.

that occur with gradient-based solvers [16, 17]. Finally, while Monte Carlo sampling is expensive, shooting methods can be effectively parallelized across processes and the advent of simulations on GPUs also provides a means of acceleration [18, 13]. However, some aspects of black-box optimization are open to criticism. Sample-based solvers such as the cross-entropy method (CEM) [19] appear 'wasteful', ignoring computation by throwing away the majority of samples, while others enforce high-entropy search distributions to avoid premature convergence [18]. Moreover, many design decisions and hyperparameters are heuristic in nature, which is undesirable from both the user- and research perspective when interpreting, tuning or advancing these methods.

In this work, we consider Monte Carlo optimal control through the broader perspective of inference-based control [20, 21, 22, 23, 24, 25, 26, 27], where optimization is achieved through importance sampling [28]. This approach covers settings such as policy search [29], motion planning [8, 30] and model predictive control (MPC) [18]. From this view point, we highlight two key design decisions: the likelihood temperature and the distribution over action sequences. An adaptive temperature scheme is crucial for controlling the optimization behavior across objectives and distributions, but in many methods this aspect is ignored or opaque. Moreover, correlated action sequences are equally crucial for performing effective exploration and control in practical settings. *Smoothness*, arising from such correlations, is an aspect of human motion [31]. Smooth priors have taken many forms across domains, such as movement primitives [32], smoothed- [11, 12] or coloured noise [4]. We use Gaussian processes [33] as action priors and show how they can be scaled to high-dimensional action spaces through factorization of the covariance. Evaluating on simulated robotic systems, we reproduce prior results on policy search while transferring these ideas to MPC, matching prior performance with respect to sample efficiency while ensuring smooth actuation.

**Contribution.** First, we present a perspective of episodic inference-based control based on Gibbs posteriors. Using this view, we then present novel Monte Carlo variants that incorporate the approximate inference error due to importance sampling, simplifying the hyperparameter while providing regularization. Thirdly, we demonstrate how richer Gaussian process priors can be combined with these regularized Gibbs posteriors for Monte Carlo MPC using online sequential inference, which achieves greater smoothness and sample efficiency than standard white noise priors. We highlight connections between this approach to MPC and effective prior approaches to episodic policy search.

## 2 Monte Carlo Methods for Optimal Control

This section outlines the problem setting and introduces variational optimization and posterior policy iteration methods. We consider the standard (stochastic) optimal control setting in discrete-time, with states $s \in \mathbb{R}^{d_s}$, actions $a \in \mathbb{R}^{d_a}$. Optimization is framed as maximizing a reward $r : \mathbb{R}^{d_s} \times \mathbb{R}^{d_a} \to \mathbb{R}$ under dynamics $p(s_{t+1} \mid s_t, a_t)$ and initial state distribution $p(s_1)$,

$$\max_{a_1, \ldots, a_T} \mathbb{E}\left[\sum_{t=1}^{T} r(s_t, a_t)\right] \quad \text{s.t.} \quad s_{t+1} \sim p(\cdot \mid s_t, a_t), \quad s_1 \sim p(s_1). \tag{1}$$

This work focuses on the episodic setting, where optimization is performed after evaluating the current solution over a finite-time horizon $T$. We frequently use the episodic return $R$, where $R(S, A) = \sum_{t=1}^{T} r(s_t, a_t)$, using upper-case to denote sequences, e.g. $A := \{a_1, \ldots, a_T\}$.

### 2.1 Variational Optimization with Gibbs Posteriors

The optimization outlined above is amenable to gradient-based solvers such as stochastic differential dynamic programming [34]. However, to aid optimization through exploration and regularization, we can consider optimizing a parametric *belief* over action sequences $q \in \mathcal{Q}$. The variational formulation (Equation 3) generalizes Bayes' rule beyond optimizing likelihoods and resembles many learning algorithms [35, 36]. This work concerns optimizing an open-loop action sequence to maximize an episodic return. Bayesian inference of an action sequence from data, known as input estimation, can be performed using message passing of the appropriate probabilistic graphical model, capturing the sequential structure of the problem and necessary priors [27]. If the measurement log-likelihood is replaced with the control objective, this inference computation can be shown to have precise dualities with dynamic programming-based optimal control [37]. While this switch in objective provides a powerful suite of inference tools for efficient computation, it requires treating the control objective as a Markovian log-likelihood, which is not the case for episodic objectives. The Gibbs likelihood is a general treatment of the objective-as-likelihood (Definition 1) [38, 39].

**Definition 1.** *(Gibbs likelihoods and posteriors) For a loss $f$ and prior $p(\boldsymbol{x})$, the Gibbs posterior $q_\alpha$ for parameter $\boldsymbol{x}$ is derived by constructing the Gibbs likelihood $\exp(-\alpha f(\boldsymbol{x}))$ from the loss,*

$$q_\alpha(\boldsymbol{x}) = \frac{1}{Z_\alpha} \exp(-\alpha f(\boldsymbol{x})) \, p(\boldsymbol{x}), \quad Z_\alpha = \int \exp(-\alpha f(\boldsymbol{x})) \, p(\boldsymbol{x}) \, \mathrm{d}\boldsymbol{x}, \quad \alpha \geq 0. \tag{2}$$

*This posterior minimizes the following objective*

$$q_\alpha = \arg\min_{q \in \mathcal{Q}} \mathbb{E}_{\boldsymbol{x} \sim q(\cdot)}[f(\boldsymbol{x})] + \tfrac{1}{\alpha} \mathbb{D}_{\mathrm{KL}}[q(\boldsymbol{x}) \,||\, p(\boldsymbol{x})]. \tag{3}$$

*This objective appears in PAC-Bayes methods [38], mirror descent methods [40] and Bayesian inference as the evidence lower bound objective when $f(\boldsymbol{x})$ is a negative log-likelihood [39].*

Augmenting the variational optimization objective with prior regularization (Equation 3), we obtain an expression of the optimal belief in the action sequence (Equation 2). The parameter $\alpha$ has a range of meanings, depending on context. In PAC-Bayes it is the dataset size, in mirror descent it is an update step size and in risk-sensitive control it is the sensitivity [41, 42]. Example 1 in Appendix A examines a tractable linear-quadratic-Gaussian example of this update, demonstrating its relation to Newton-like optimization and highlighting the effect $\alpha$ has on the regularized update.

## 2.2 Posterior Policy Iteration

The optimal control problem (Equation 1) is ambiguous regarding whether the action sequence or state-action trajectory is the optimization variable. Applying the Gibbs posterior to the optimal control setting recovers Rawlik et al.'s posterior policy iteration [41], which can be implemented using the joint distribution or policy. We consider the following joint state-action distribution, that factorizes in the following Markovian fashion $p(\boldsymbol{S}, \boldsymbol{A}) = p(\boldsymbol{s}_1) \prod_{t=1}^{T} p(\boldsymbol{s}_{t+1} \mid \boldsymbol{s}_t, \boldsymbol{a}_t) \, p(\boldsymbol{a}_t \mid \boldsymbol{s}_t)$. Posterior policy iteration updates the state-action distribution through the policy, constructing a Gibbs likelihood from the reward, as the dynamics and initial state distribution are constant.

**Definition 2.** *(Posterior policy iterations (PPI) [43]) As the initial distribution and dynamics are shared by the prior and posterior joint state-action distribution, the joint Gibbs posterior $q_\alpha(\boldsymbol{S}, \boldsymbol{A}) \propto \exp(\alpha R(\boldsymbol{S}, \boldsymbol{A})) \, p(\boldsymbol{S}, \boldsymbol{A})$ can be alternatively expressed using the policy posterior update $q_\alpha(\boldsymbol{A} \mid \boldsymbol{S}) \propto \exp(\alpha R(\boldsymbol{S}, \boldsymbol{A})) \, p(\boldsymbol{A} \mid \boldsymbol{S})$.*

Using this update, the key decisions are choosing $p(\boldsymbol{A} \mid \boldsymbol{S})$, $\alpha$ and the inference approximation. If $p$ and $q_\alpha$ are Gaussian, then PPI involves iterative refinement of the distribution. In the Monte Carlo setting, $q_\alpha$ takes the form of an importance-weighted empirical distribution. To apply iteratively, $p$ is updated using the M-projection, following the objective (Equation 3), i.e. a weighted maximum likelihood fit of the policy parameters [29]. This approach is a stochastic approximate expectation maximization (SAEM) method [44] and described fully in Algorithm 1 in the Appendix. We argue a key aspect of PPI methods is how to specify the inverse temperature $\alpha$ during optimization (Section 3), as it has a strong influence on the posterior, which is important when fitting rich distributions such as Gaussian processes (Section 4) from samples. Gaussian process action priors can be applied to several control settings, such as policy search and model predictive control (Section 6).

## 3 Posterior Policy Constraints for Monte Carlo Optimization

The Gibbs posterior in Definition 2 has been adopted widely in control, albeit from a range of different perspectives, such as Bayesian smoothing [23], solutions to the Feynman-Kac equation [45], maximum entropy [26], mirror descent [46] and entropy-regularized reinforcement learning [47]. An open question is how best to set $\alpha$ for Monte Carlo optimization? Relative entropy policy search (Definition 3), provides a principled and effective means of deriving $\alpha$ for stochastic optimization, using the constrained optimization view of entropy-regularized optimal control.

**Definition 3.** *(Episodic relative entropy policy search (eREPS) [29]) Maximize the expected return, subject to a hard KL bound $\epsilon$ on the policy update,*

$$\max_{\boldsymbol{\theta}} \mathbb{E}_{\boldsymbol{s}_{t+1} \sim p(\cdot|\boldsymbol{s}_t, \boldsymbol{a}_t), \boldsymbol{a}_t \sim q_{\boldsymbol{\theta}}(\cdot|\boldsymbol{s}_t), \boldsymbol{s}_1 \sim p(\cdot)}[R(\boldsymbol{s}_t, \boldsymbol{a}_t)] \quad s.t. \quad \mathbb{D}_{\mathrm{KL}}[q_{\boldsymbol{\theta}}(\boldsymbol{A}|\boldsymbol{S}) \,||\, p(\boldsymbol{A}|\boldsymbol{S})] \leq \epsilon.$$

*The posterior policy takes the form $q_{\boldsymbol{\theta}}(\boldsymbol{A}|\boldsymbol{S}) \propto \exp(\alpha R) \, p(\boldsymbol{A}|\boldsymbol{S})$, where $\alpha$ is derived from Lagrange multiplier calculated by minimizing the empirical dual $\mathcal{G}(\cdot)$ using $N$ samples,*

$$\min_\alpha \mathcal{G}(\alpha) = \frac{\epsilon}{\alpha} + \frac{1}{\alpha} \log \int p(\boldsymbol{S}, \boldsymbol{A}) \exp(\alpha R(\boldsymbol{S}, \boldsymbol{A})) \, \mathrm{d}\boldsymbol{S} \, \mathrm{d}\boldsymbol{A} \approx \frac{\epsilon}{\alpha} + \frac{1}{\alpha} \log \frac{1}{N} \sum_{n=1}^{N} \exp(\alpha R_n).$$

While REPS is a principled approach to stochastic optimization, we posit two weaknesses: The hard KL constraint $\epsilon$ is difficult to specify, as it depends on the optimization problem, distribution family and dimensionality. Secondly, the Monte Carlo approximation of the dual has no regularization and may poorly adhere to the KL constraint without sufficient samples. Therefore, we desire an alternative approach that resolves these two issues, capturing the Monte Carlo approximation error with a simpler hyperparameter. To tackle this problem, we interpret the REPS update as a pseudo-posterior, where the temperature is calculated using the KL constraint. We make this interpretation concrete by reversing the objective and constraint, switching to an equality constraint for the expectation,

$$\min_{\boldsymbol{\theta}} \mathbb{D}_{\mathrm{KL}}[q_{\boldsymbol{\theta}}(\boldsymbol{A} \mid \boldsymbol{S}) \mid\mid p(\boldsymbol{A} \mid \boldsymbol{S})] \quad \text{s.t.} \quad \mathbb{E}_{\boldsymbol{s}_{t+1} \sim p(\cdot|\boldsymbol{s}_t, \boldsymbol{a}_t),\, \boldsymbol{a}_t \sim q_{\boldsymbol{\theta}}(\cdot|\boldsymbol{s}_t),\, \boldsymbol{s}_1 \sim p(\cdot)}[\textstyle\sum_t r(\boldsymbol{s}_t, \boldsymbol{a}_t)] = R^*.$$

This objective is a *minimum relative entropy problem* [48], which yields the same Gibbs posterior as eREPS (Lemma 1, Appendix A). With exact inference, a suitable prior and oracle knowledge of the maximum return, this program computes the optimal policy in a single step by setting $R^*$ to the optimal value. However, in this work, the expectation constraint requires self-normalized importance sampling (SNIS) on sampled returns $R^{(n)}$ using samples from the current policy prior,

$$\mathbb{E}_{\boldsymbol{s}_{t+1} \sim p(\cdot|\boldsymbol{s}_t, \boldsymbol{a}_t),\, \boldsymbol{a}_t \sim q_{\boldsymbol{\theta}}(\cdot|\boldsymbol{s}_t),\, \boldsymbol{s}_1 \sim p(\cdot)}[\textstyle\sum_t r(\boldsymbol{s}_t, \boldsymbol{a}_t)] \approx \textstyle\sum_n w_{q/p}^{(n)} R^{(n)} = \frac{\sum_n R^{(n)} \exp(\alpha R^{(n)})}{\sum_n \exp(\alpha R^{(n)})} = R^*.$$

Rather than specifying $R^*$ here, we identify that this estimator is fundamentally limited by inference accuracy. We capture this error by applying an IS-derived concentration inequality to this estimate (Theorem 1) [49]. This lower bound can be used as an objective for optimizing $\alpha$, balancing policy improvement with approximate inference accuracy.

**Theorem 1.** *(Importance sampling estimator concentration inequality (Theorem 2, [49])) Let $q$ and $p$ be two probability densities such that $q \ll p$ and $d_2[q \mid\mid p] < +\infty$. Let $\boldsymbol{x}_1, \boldsymbol{x}_2, \ldots, \boldsymbol{x}_N$ i.i.d. random variables sampled from $p$ and $f : \mathcal{X} \to \mathbb{R}$ be a bounded function $(||f||_\infty < +\infty)$. Then, for any $0 < \delta \le 1$ and $N > 0$ with probability at least $1 - \delta$:*

$$\mathbb{E}_{\boldsymbol{x} \sim q(\cdot)}[f(\boldsymbol{x})] \ge \frac{1}{N} \sum_{i=1}^{N} w_{q/p}(\boldsymbol{x}_i) f(\boldsymbol{x}_i) - ||f||_\infty \sqrt{\frac{(1 - \delta) d_2[q(\boldsymbol{x}) \mid\mid p(\boldsymbol{x})]}{\delta N}}. \tag{4}$$

The divergence term $d_2[q \mid\mid p]$ is the exponentiated Rényi-2 divergence, $\exp \mathbb{D}_2[q \mid\mid p]$. While this is tractable for the multivariate Gaussian, it is otherwise not available in closed form. Fortunately, we can use the effective sample size (ESS) [50] as an approximation, as $\hat{N}_\alpha \approx N / d_2[q_\alpha \mid\mid p]$ [49, 51] (Lemma 2, see Section A of the Appendix). Combining Equation 4 with our constraint, instead of setting $R^*$, we maximize the IS lower bound $R_{\mathrm{LB}}^*$ to form an objective for the inverse temperature $\alpha$ which incorporates the inference accuracy due to the sampling given inequality probability $1 - \delta$,

$$\max_{\alpha} R_{\mathrm{LB}}^*(\alpha, \delta) = \mathbb{E}_{q_\alpha/p}[R] - \mathcal{E}_R(\delta, \hat{N}_\alpha), \quad \mathcal{E}_R(\delta, \hat{N}_\alpha) = ||R||_\infty \sqrt{\frac{(1 - \delta)}{\delta}} \frac{1}{\sqrt{\hat{N}_\alpha}}. \tag{5}$$

We refer to this approach as *lower-bound policy search* (LBPS). This objective combines the expected performance of $q_\alpha$, based on the IS estimate $\mathbb{E}_{q_\alpha/p}[\cdot]$, with regularization $\mathcal{E}_R$ based on the return and inference accuracy. Treating $p$, $N$, $||R||_\infty$ as task-specific hyperparameters, the only algorithm hyperparameter $\delta \in [0, 1)$ defines the probability of the bound. In practice, self-normalized importance sampling is used for PPI, as the normalizing constants of the Gibbs likelihoods are not available. While Metelli et al. also derive an SNIS lower bound [49], we found, as they did, that the IS lower bound with SNIS estimates work better in practice due to the conservatism of the SNIS bound. An interpretation of this approach is that the Rényi-2 regularization constrains the Gibbs posterior to be one that can be estimated from the finite samples, as the divergence is used in evaluating IS sample complexity [52, 53]. Moreover, the role of the ESS for regularization is similar to the 'elite' samples in CEM. Connecting these two mechanisms as robust maximum estimators (Section A), we also propose *effective sample size policy search* (ESSPS), which optimizes $\alpha$ to achieve a desired ESS $N^*$, i.e. a Rényi-2 divergence bound, using the objective $\min_\alpha |\hat{N}_\alpha - N^*|$. More details regarding PPI (Section A) and temperature selection methods (Table 1) are in the Appendix.

This section introduces two methods, LBPS and ESSPS, for constraining the Gibbs posteriors for Monte Carlo optimization. These methods provide statistical regularization through soft and hard constraints involving the effective sample size, which avoids the pitfall of fitting high-dimensional distributions to a few effective samples. A popular setting for these methods is MPC, which performs episodic optimization over short planning horizons while adapting each time step to the current state. Moreover, for optimal control, we also need to specify a suitable prior over action sequences. To apply PPI to the MPC setting, we must implement online optimization given this prior over actions.

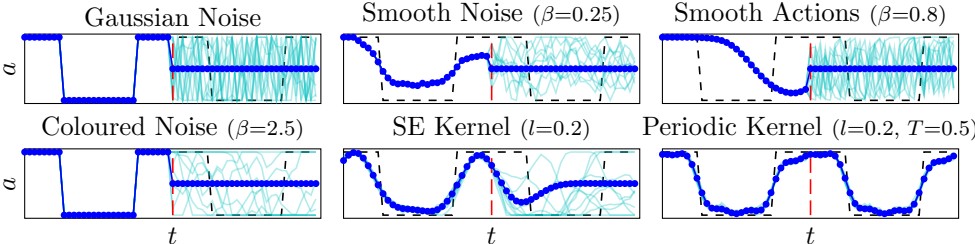

Figure 2: A practical aspect of Monte Carlo control methods for robotics is optimizing smooth action sequence. This example shows a non-smooth optimal sequence - - -, which may be undesirable, though optimal, to fit exactly. Prior methods struggle at providing both effective smooth solutions in the mean —— and action samples ——, as they ultimately fit the action distribution in an independent fashion. Using kernel-derived covariance function provides both. The line - - - denotes the optimization horizon, beyond which is exploratory actions derived from both the posterior and prior.

## 4  Online Posterior Policy Iteration & Prior Design

In this section, we derive the online realization of posterior policy iteration that uses and maintains correlated action priors, computing the finite-horizon $H$ future actions given a likelihood on a subset of actions from the past. $R$ represents the return-based Gibbs likelihood term (Definition 1),

$$q_\alpha(\boldsymbol{a}_{t:t+H} \mid R_{1:\tau}) = \int q_\alpha(\boldsymbol{a}_{1:t+H} \mid R_{1:\tau})\,\mathrm{d}\boldsymbol{a}_{1:t-1} \propto \int p(R_{1:\tau} \mid \boldsymbol{a}_{1:\tau})\,p(\boldsymbol{a}_{1:t+H})\,\mathrm{d}\boldsymbol{a}_{1:t-1}, \quad (6)$$

where $\tau \le t + H$. As an analogy, this is equivalent to combining forecasting with state estimation, i.e. $p(\boldsymbol{x}_{t:t+H}|\boldsymbol{y}_{1:t})$ for states $\boldsymbol{x}$ and measurements $\boldsymbol{y}$. For correlated priors on the action space, this computation is tractable if working with Gaussian processes. In fact, a recurring aspect across several many posterior policy iteration-like approaches is the use of Gaussian process policies,

$$p(\boldsymbol{A} \mid \boldsymbol{S}) = \begin{cases} \prod_t \mathcal{N}(\boldsymbol{\mu}_t, \boldsymbol{\Sigma}_t), & \text{(Independent Gaussian noise, e.g. [18]),} \\ \mathcal{N}(\boldsymbol{\mu}_{\boldsymbol{w}}^\top \phi(t), \phi(t)^\top \boldsymbol{\Sigma}_{\boldsymbol{w}} \phi(t)), & \text{(Bayesian linear regression, e.g. ProMP [32]),} \\ \prod_t \mathcal{N}(\boldsymbol{k}_t + \boldsymbol{K}_t \boldsymbol{s}, \boldsymbol{\Sigma}_t), & \text{(time-varying linear Gaussian e.g. [41, 54, 42]),} \\ \mathcal{GP}(\boldsymbol{\mu}(\boldsymbol{s}), \boldsymbol{\Sigma}(\boldsymbol{s})), & \text{(non-parametric Gaussian process [55]).} \end{cases}$$

Despite the simplicity of Gaussian action noise, for robotics, more sophisticated noise is often desired for safety and effective exploration [56, 29]. Prior work has proposed first-order smoothing [11, 12]. Using $\boldsymbol{v}_t^{(n)} \sim \mathcal{N}(\boldsymbol{0}, \boldsymbol{I})$, $\beta \in [0, 1]$ and $\boldsymbol{\Sigma}_t = \boldsymbol{L}_t \boldsymbol{L}_t^\top$, actions are sampled using

$$\boldsymbol{a}_t^{(n)} = \boldsymbol{\mu}_t + \boldsymbol{L}_t \boldsymbol{n}_t^{(n)}, \quad \boldsymbol{n}_t^{(n)} = \beta \boldsymbol{v}_t^{(n)} + (1-\beta)\boldsymbol{n}_{t-1}^{(n)}, \text{ or } \boldsymbol{n}_t^{(n)} = \beta \boldsymbol{v}_t^{(n)} + \sqrt{(1-\beta^2)}\,\boldsymbol{n}_{t-1}^{(n)}.$$

However, in practice it is also implemented as $\boldsymbol{a}_t^{(n)} = \beta(\boldsymbol{\mu}_t + \boldsymbol{L}_t \boldsymbol{v}_t^{(n)}) + (1-\beta)\boldsymbol{a}_{t-1}^{(n)}$ [1]. While this approach directly smooths the actuation, it also introduces a lag, which may deteriorate performance. Other approaches have used colored noise for sampling the noise $\boldsymbol{n}$ [4]. Contrast these approaches to Gibbs sampling a multivariate Gaussian joint distribution with 1-step cross-correlations [58], which is $\boldsymbol{a}_{t|t-1}^{(n)} = \boldsymbol{\mu}_{t|t-1}^{(n)} + \boldsymbol{L}_{t|t-1}\boldsymbol{v}_t^{(n)}$, where $\boldsymbol{\mu}_{t|t-1}^{(n)} = \boldsymbol{\mu}_t + \boldsymbol{\Sigma}_{t,t-1}\boldsymbol{\Sigma}_{t-1}^{-1}(\boldsymbol{a}_{t-1}^{(n)} - \boldsymbol{\mu}_{t-1})$, and $\boldsymbol{\Sigma}_{t|t-1} = \boldsymbol{\Sigma}_t - \boldsymbol{\Sigma}_{t,t-1}\boldsymbol{\Sigma}_{t-1}^{-1}\boldsymbol{\Sigma}_{t,t-1}^\top$. The differences are subtle, but important. The initial proposed sampling scheme essentially adds correlated noise to the mean for exploration, but does not consider the smoothness of the mean itself. The practical implementation incorporates the previous action, but through exponential smoothing, which introduces a fixed lag that potentially degrades the quality of the mean action sequence. Correct sampling of the joint distribution has neither of these issues and naturally extends to correlations over several time steps. We do this in a general fashion by considering the (continuous time) Gaussian process (see Section G, Appendix), so $p(\boldsymbol{a}_t) = \mathcal{N}(\boldsymbol{\mu}_{t_i:t_j}, \boldsymbol{\Sigma}_{t_i:t_j}) = \mathcal{GP}(\boldsymbol{\mu}(\boldsymbol{t}), \boldsymbol{\Sigma}(\boldsymbol{t}))$ for a discrete-time sequence $\boldsymbol{t} = [t_i, \ldots, t_j]$. Proposition 1 shows how the time shift for MPC can be implemented in a general fashion when using GPs.

**Proposition 1.** *Given a Gaussian process prior $\mathcal{GP}(\boldsymbol{\mu}(t), \boldsymbol{\Sigma}(t))$ and multivariate normal posterior $q_\alpha(\boldsymbol{a}_{t_1:t_2}) = \mathcal{N}(\boldsymbol{\mu}_{t_1:t_2|R}, \boldsymbol{\Sigma}_{t_1:t_2|R})$ for $t_1$ to $t_2$, the posterior for $t_3$ to $t_4$ is expressed as*

$$\boldsymbol{\mu}_{t_3:t_4|R} = \boldsymbol{\mu}_{t_3:t_4} + \boldsymbol{\Sigma}_{t_3:t_4,t_1:t_2}\boldsymbol{\nu}_{t_1:t_2}, \quad \boldsymbol{\Sigma}_{t_3:t_4|R} = \boldsymbol{\Sigma}_{t_3:t_4} - \boldsymbol{\Sigma}_{t_3:t_4,t_1:t_2}\boldsymbol{\Lambda}_{t_1:t_2}\boldsymbol{\Sigma}_{t_3:t_4,t_1:t_2}^\top, \quad (7)$$

*where $\boldsymbol{\nu}_{t_1:t_2} = \boldsymbol{\Sigma}_{t_1:t_2}^{-1}(\boldsymbol{\mu}_{t_1:t_2|R} - \boldsymbol{\mu}_{t_1:t_2})$ and $\boldsymbol{\Lambda}_{t_1:t_2} = \boldsymbol{\Sigma}_{t_1:t_2}^{-1}(\boldsymbol{\Sigma}_{t_1:t_2} - \boldsymbol{\Sigma}_{t_1:t_2|R})\boldsymbol{\Sigma}_{t_1:t_2}^{-1}$.*

This update combines the new sequence prior from $t_3$ to $t_4$ and the previous likelihood used in the update for $t_1$ to $t_2$, obtained from the posterior and prior. Note, the cross-covariance $\boldsymbol{\Sigma}_{t_3:t_4,t_1:t_2}$ is computed using the covariance function of the prior GP. The proof is in Appendix A.

---

[1]See the source code for Nagabandi et al. [11] and `MBRL-lib` [57].

For a stationary kernel, fixed planning horizon and fixed control frequency, the term $\boldsymbol{\Sigma}_{t_1:t_2}^{-1}$ is $\boldsymbol{\Sigma}_{t:t+H}^{-1}$ and is constant, so can be computed at initialization to avoid repeated inversion. Figure 2 demonstrates how this update lets us combine our prior with previous posterior in a principled fashion. Moreover, its continuous-time construction means that the time resolution can be updated, not just the time window, for planning at different timescales [30].

Compared to the independence assumption, modeling correlations between actions introduces complexity. The full covariance over (flattened) time and action has a complexity $R(T^3 d_a^3)$, which is infeasible to work with. Assuming independence between actions, a GP per action has a complexity of $R(T^3 d_a)$, requiring $d_a$ GPs to be fit, which is not desirable for online methods such as MPC. To avoid the linear scaling w.r.t. $d_a$, we propose using the *matrix Normal distribution* (Definition 4) for scalability, as it is parameterized into single $T$ and $d_a$-dimensional covariances,

**Definition 4.** *(Matrix Normal Distribution (MaVN) [59]) For a random matrix $\boldsymbol{X} \in \mathbb{R}^{n \times p}$, it follows the distribution $\boldsymbol{X} \sim \mathcal{MN}(\boldsymbol{M}, \boldsymbol{K}, \boldsymbol{\Sigma})$, where $\boldsymbol{M} \in \mathbb{R}^{n \times p}$, $\boldsymbol{K} \in \boldsymbol{S}_+^n$ and $\boldsymbol{\Sigma} \in \boldsymbol{S}_+^p$, if and only if $vec(\boldsymbol{X}) \sim \mathcal{N}(vec(\boldsymbol{M}), \boldsymbol{\Sigma} \otimes \boldsymbol{K})$, for Kronecker product $\otimes$ and $\boldsymbol{S}_+^k = \{\boldsymbol{X} \in \mathbb{R}^{k \times k} | \boldsymbol{X}^\top = \boldsymbol{X}, \boldsymbol{X} \succeq 0\}$.*

Using the Kronecker-structured covariance provides a useful decomposition of the time-based covariance $\boldsymbol{K}$, that defines correlations between time steps, and an action covariance $\boldsymbol{\Sigma}$ that captures correlations between actions. Typically we assume actions are independent, but cross-correlations could be learned from experience for richer coordination. While this Kronecker structure does not fully capture the correlations between time and actions, the structure is very useful for MPC on robotic systems, where the actions space could be very high but the planning horizon is sufficiently small for covariance estimation using a reasonable number of Monte Carlo rollouts.

**Feature Approximations.** Despite the matrix Normal factorization, computing the correlations between actions still requires a dense $H \times H$ covariance matrix $\boldsymbol{K}$ for planning horizon $H$. To sparsify this quantity, we consider kernel approximations, such as the canonical basis functions $\sum_n k(\cdot, \boldsymbol{x}_n)$ and *spectral* approximations using random features $\sum_n \phi_n(\cdot)$ [60], for a Bayesian linear model $\boldsymbol{\phi}_t^\top \boldsymbol{W}$. Focusing on the squared exponential (SE) kernel, this results in radial basis function (RBF) and random Fourier features (RFF) respectively. Interestingly, RBF features are closely related to probabilistic movement primitives, used extensively in policy search for robotics [32]. For one-dimensional inputs, RFFs are effectively approximated by applying Gauss-Hermite quadrature [61] to the random weights [62]. RBF features and RFFs approximate w.r.t. time and frequency respectively and could be combined [63]. Using these continuous-time features, the optimization is now abstracted from planning horizon and control frequency, providing much greater flexibility. Secondly, due to the features, a factorized weight covariance approximation does not sacrifice smoothness. Moreover, the moment updates described above are not needed, as only $\boldsymbol{\phi}_t$ is updated.

## 5    Related Work

**Inference-based control.** Posterior policy iteration was proposed by Rawlik et al. [41] and covers prior methods developed from Bayesian smoothing [23, 37], expectation maximization [22, 56], entropy regularization [47, 9] and path integral [64] perspectives. For MPC specifically, the path integral-based MPPI was proposed [18], with alternative formulations based on mirror descent [46] and variational inference [5, 65, 66]. Mukadam et al. [25] models the optimal state-action distribution as a sparse Gaussian process and uses linearization for approximate inference. The same approach is used for Gaussian process motion planning [30], which are also optimized using sampling [8]. Gaussian quadrature is also used for inference-based MPC [27]. Concurrent work uses the ESS for a temperature adjusting heuristic for MPPI [15] and also combines policy search with MPC using PPI techniques [67]. See Section B for a more in-depth discussion on these related works.

**Policy design and regularization.** Smooth actuation is important in robot learning for safety and exploration, having been proposed for Monte Carlo MPC [11, 12, 4] and more broadly incorporated using augmented objectives, parameter sampling and policy design, e.g. [68, 69, 70].

**Stochastic search.** Probabilistic interpretations of black-box optimization algorithms are well established [71, 72, 73], however prior work did not connect the ESS and elite samples. CEM and extensions have also been adopted widely as a solver for MPC [3, 4, 6].

**Gaussian processes for control.** This work adopts GPs for correlated action priors. This is distinct from prior work which uses GPs to approximate dynamics or value functions, e.g. [74, 75, 76, 77].

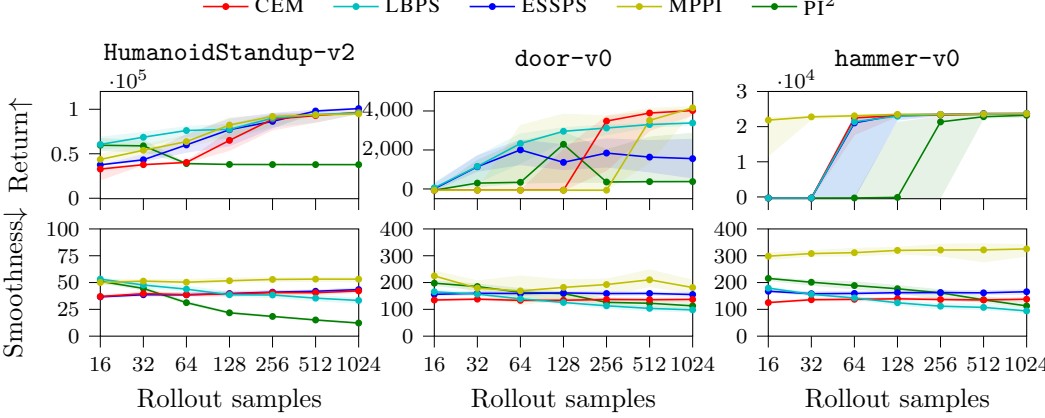

Figure 3: MPC return and smoothness with white noise priors. Displaying quartiles over 50 seeds. These priors require a large number of samples for good performance, across methods and tasks.

## 6 Experimental Results

We assess the Gibbs posterior methods and policy design empirically across various settings. Black-box optimization (Section 6.1) considers standard benchmarks, while policy search (Section 6.2) optimizes action sequences for a robotic task. For MPC (Section 6.3), we evaluate online PPI approaches with white noise and smooth priors on high-dimensional, contact-rich tasks. For the code, see github.com/JoeMWatson/monte-carlo-posterior-policy-iteration.

### 6.1 Black-box Optimization

To understand the behaviour of the proposed PPI variants, the performance of LBPS and ESSPS on a range of standard black-box optimization functions over a range of hyperparameters, with eREPS and CEM as baselines, are shown in Appendix D.1. Figures 7 – 10 show that ESS is a useful metric for these methods, as each solver exhibits consistent ESS for a given hyperparameter value. However, the uniform weights used by CEM (Figure 7) maintain entropy longer than ESSPS (Figure 8), which can lead to better optima, so the ESS is not sufficient to fully capture the behavior of these solvers.

### 6.2 Policy Search

As LBPS and ESSPS are closely related to eREPS, we repeat the experiment from prior work performing the 'ball in a cup' task using policy search using a Barret WAM [14], which has been shown to transfer to the physical system [78, 14, 79], as a benchmark task. Moreover, we replace ProMPs with Matrix normal RBF and RFF policies. From the kernel perspective, this feature approximation is motivated by the large ($T \simeq 1000$) task horizon. The results in Appendix D.2 confirm that these solvers are all capable of solving the task, based on success rate, where RBF (Figure 11) and RFF features (Figure 12) perform equally well w.r.t. the convergence of the success rate for each approach.

### 6.3 Model Predictive Control with Oracle Dynamics

We evaluate online PPI across a range of high-dimensional robotic control tasks in MuJoCo [80], including HumanoidStandup-v2 in Gym [81] and door-v0, hammer-v0 from mj_envs, using the Adroit hand (Figure 1) [82]. To measure smoothness, we adopt the FFT-based score $\frac{2}{Nf_s} \sum_{i=1}^{N} a_i f_i$ [68], with sampling frequency $f_s$ and $N$ resolvable frequencies $f$ with amplitudes $a$. We compute the Euclidean norm of the action sequence over time and apply the smoothness measure to this signal. For the evaluation, we focus on a low computational budget, with 1 or 2 iterations per timestep. To assess the impact of approximate inference, we assess performance over an logarithmic range of sample rollouts, following prior work [4]. Details may be found in Appendix E.2.

**White noise priors.** Figure 3 shows MPC with white noise priors using LBPS and ESSPS, with MPPI, CEM and PI$^2$ baselines (see Table 1). While each solver performs comparably for 1024 rollouts, the low sample regime shows greater performance variance. While MPPI seems particularly effective, Figure 13 shows that its average ESS is particularly low, $\simeq 1$ for many cases. Combined with the fixed variances, this suggests optimization is closer to greedy random search than importance sampling. The poor door-v0 performance of ESSPS is due to slow opening, rather than task failure.

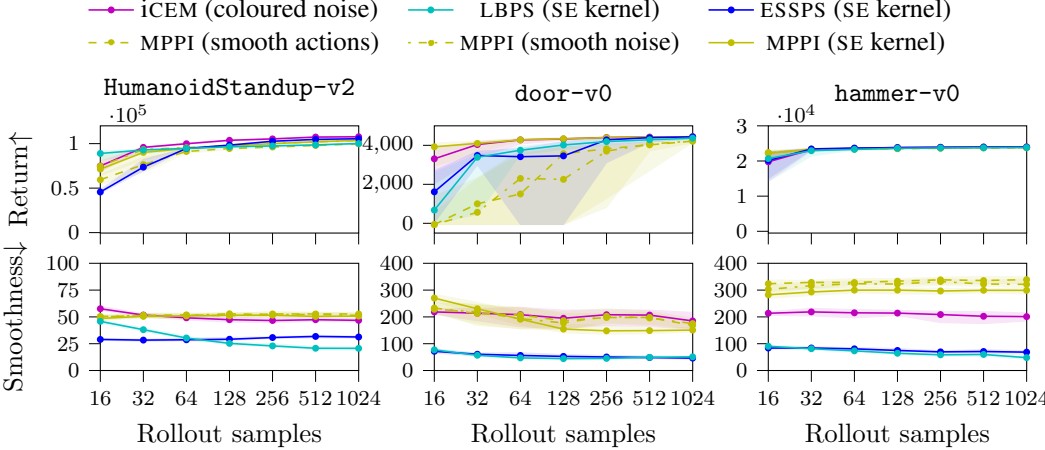

Figure 4: MPC return and smoothness with smooth action priors. Displaying quartiles over 50 seeds. Compared to white noise priors, smooth action priors improve sample efficiency dramatically, but only PPI methods (LBPS, ESSPS) preserve this smoothness in the downstream control.

**Policy design for smooth control.** Figure 4 shows online PPI with action priors. LBPS, ESSPS and MPPI use the SE kernel, with iCEM [4] and MPPI with smooth actions and noise as a baseline. The smooth action distributions greatly improves performance across models, tasks and sample sizes, due to effective exploration. As desired, the richer SE kernel provides much greater smoothness, by up to a factor of 2 compared to baselines, with limited impact to performance. It is unsurprising that the smoothness bias reduces performance if optimal behavior is non-smooth, as illustrated in Figure 2. Appendix D.3.2 shows some of the action sequences from Figure 4, where we see GP smoothness varies with increasing rollout samples and also results in significant actuator amplitude reduction.

**Comparing kernel- and feature-based policies.** To assess feature approximations for smooth MPC, we replace the SE kernel with RBF and RFF features, while keeping the lengthscale fixed. These policies perform worse given fewer samples, but are comparable to the true kernel with sufficient samples (Figure 15). We attribute this to the compounding errors of kernel approximation and fewer effective samples. In contrast to the policy search task, RFFs appear superior to RBF features.

**Learning priors from data.** A benefit of using GP priors is the ability to optimize hyperparameters from expert demonstrations through the likelihood or a divergence. Moreover, the matrix normal distribution is useful for analyzing high-dimensional action sequences, as it decomposes temporal- and action correlations into viewable covariance matrices. Section D.3.3 shows the matrix normal distributions of expert demonstrations of the tasks, obtained through human and RL experts. The results show that, surprisingly, the demonstrations are rougher than the smoothness achievable with MPC. We attribute this to control artifacts from demonstration collection and the use of Gaussian noise by RL agents. Applying the same methodology to the demonstrations of the smooth MPC agents proposed here extracts the expected action correlations across tasks. This analysis also raises the question of whether smoothness is an inductive bias we enforce for practicality, or a phenomena we expect to arise from optimality. If the latter, it may be that the simulated environments or objectives considered are lacking components that encourage smoothness, such as energy efficiency.

## 7 Conclusion

We present a broad perspective on episodic posterior policy iteration method for robotics and new methods for the Monte Carlo setting, based on regularizing the IS approximations. By considering vector-valued Gaussian processes for action priors, we have demonstrated how sample-efficient MPC can be performed as online inference and with greater control over actuator smoothness, connecting Monte Carlo MPC to prior work on policy search. This approach was validated on a set of high-dimensional MPC tasks closely matching baseline performance while achieving greater smoothness.

**Limitations.** Much of the prior work is motivated by simplicity, minimizing hyperparameter tuning and numerical procedures such as matrix inversion [64]. In contrast, the contributions of this work introduces complexity, i.e. online temperature optimization and the use of dense covariance matrices in order to perform more sophisticated approximate inference. While this additional complexity has an impact on execution time (Table 2, Appendix), we hope the sample-efficiency when combined with accelerations such as GPU-integration should produce real-time algorithms [13].

**Acknowledgments**

This work built on prior codebases developed by Johannes Silberbauer, Michael Lutter and Hany Abdulsamad. The large-scale experiments and ablations were conducted on the Lichtenberg high performance computer of the TU Darmstadt. The authors wish to thank Hany Abdulsamad, Boris Belousov, Michael Lutter, Fabio Muratore, Pascal Klink, Georgia Chalvatzaki, Kay Hansel, Oleg Arenz and the anonymous conference reviewers for helpful feedback on earlier drafts.

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
