# OpenReview forum: "Inferring Smooth Control: Monte Carlo Posterior Policy Iteration with Gaussian Processes"
_robot-learning.org/CoRL/2022/Conference — CoRL 2022 Oral_

### Official Review · Reviewer_9hau · 2022-07-28

**Originality:** Very Good
**Technical Quality:** Excellent
**Clarity Of Presentation:** Excellent
**Impact:** 3

**Recommendation:**

Strong Accept: I recommend accepting the paper and will argue for my recommendation even if other reviewers hold a different opinion.

**Summary:**

The paper proposes a novel way to perform inference-based control with Monte Carlo sampling methods in online sequential inference using Gaussian Process priors to promote smoothness in both action priors and samples. The novelty lies in how the action distribution is factorized and the introduction of a theoretically grounded approach to optimize the likelihood temperature.

The relationship of the proposed method and prior work, such as CEM and MPPI is well presented and explored. In fact, sections 2-4 describing the background and method are laid out very clearly and with a very solid theoretical setting. Prior work is reframed in the same statistical interpretation and provide an insightful interpretation of their connection to the current work. Additionally, the paper considers a extensive number of relevant previous works, close to a literature review on sampling-based control as inference.


**Issues:**

I have the following minor corrections and/or recommendations:
- In line 97 "from" should read "form";
- In line 129 "Reny" should read "Renyi";
- In the equation in line 154, I believe the first integral term should be the
  joint distribution and not the marginal;
- The claim in line 159 warrants a citation;
- Reference [13] points to a paper published on CoRL 2022;
- Please identify what the parameter `k` refers to on Figures 5 and 6;
- On Figures 9 and 10 are rollouts referring to the cumulative sampled rollouts
  or episodes?

**Quality Of The Limitations Section:**

Additional details required

**Reviewer Expertise:**

4: The reviewer is confident but not absolutely certain that the evaluation is correct

**Robotics Focus:**

Highly relevant to robotics but no hardware experiments

**Strengths And Weaknesses:**

The paper is very well written and has extensive background research. The content is delivered in a clear and logical progression and there are several simulated results. The contribution is also relevant and although both LBPS and ESSPS still have hyperparemeters to control how the temperature is optimized (respectively the probability of the bound and effective sample size) these offer a more intuitive selection. Lastly, the authors present a feasible implementation and address how the proposed method could be efficiently implemented for online use.

On the other hand, I feel the motivation for the desirability of smooth priors and actions could be stronger. At passages there seems to be a lack of distinction between correlation and smoothness, and while correlation is important for a coordinated exploration of the sampling space, it was not clear or well referenced when smooth controls are necessary. In fact, it is known that for minimum-time control problems where the Hamiltonian is linear w.r.t. the control variable, the optimal solution will be a bang-bang controller and many lower-level actuators already cope with abrupt changes in the control signal. Furthermore, on some of the simulation tasks, e.g. black-box optimization, there seems to be a gap in performance which is not attributable to feature approximations and it would be valuable to have a possible explanation from the authors.


**Summary Of Recommendation:**

I strongly recommend that the paper be accepted for oral presentation. Despite my criticism on the motivation above, the contributions are clear and relevant and the theoretical framework is very well presented. The method is validated through simulation results only, but is similar enough in nature as to accept that it would be extendable to hardware where the baselines have been applied before. This is further emphasized when considering the computational performance comparison listed on supplementary material.

---

> ### Author Response · Authors · 2022-08-23
> **Rebuttal response to Reviewer 9hau**
>
> We wish to thank the reviewer for their helpful comments and kind words. We have been able to address all comments in the revision.
>
> **Smoothness.** Thanks for raising this aspect. For the most part, smooth controls are a practical desiderata for robotics (e.g. robot safety, actuator noise, energy consumption) and not included in the task objective. Moreover, we were inspired by the result of Pinneri et al. demonstrating Monte Carlo efficiency with smooth priors.  We have revised several passages to make this claim clearer, and added references where necessary. Figure 2 is indeed designed to show that there is a trade-off between optimality and smoothness if the optimal solution is non-smooth. Regarding motivation for smoothness, this work was motivated by the recurring implementation detail in sample-based MPC to smooth the action sequence heuristicly in various different ways. In this work, we make smoothness a focus of the method by considering smooth Gaussian process priors and using the posterior inference interpretation. One interesting and surprising result of this work was how enforcing smoothness results in significant (and unexpected) amplitude reduction (e.g. FIgure 1) which relates to greater energy efficiency.
>
> **Black-box optimization.** The difference between CEM and PPI methods is a good question to raise. Our hypothesis at the start of this project was that CEM and PPI / REPS methods could be unified through the effective sample size interpretation, along with entropy and the KL update step. The black-box optimization results show that this is not completely the case, in particular for the Styblinksi test function. The reason seems to be that the ‘hard’ weights of CEM make a significant difference when performing optimization. It appears the exponential weights are more conservative. Viewing the 2D Styblinksi function, it has many flat regions, so overly conservative updates may explain the local minima. We believe an interesting direction may be combining LBPS and ESSPS with the Tsallis divergence [1], which enables low-performance samples to be ignored like in CEM, which may help alleviate the conservative updates. We discuss the difference between the CEM vs ESSPS performance in Section 6.1.
>
> **Equation 6.** Thanks for highlighting this confusion. The first integral is the marginalization of the ‘old’ timesteps $1:t-1$ from the joint posterior over all timesteps $1:t+H$. The second integral replaces the posterior with the likelihood and prior according to Bayes’ rule, which could be written as a joint distribution $p(R_{1:\tau}, a_{1:t+H})$. However, as $R$ represents a Gibbs likelihood rather than a random variable, we believe this joint distribution could be a source of confusion, and therefore prefer to keep to expressions w.r.t. likelihoods, priors and posteriors like in Definition 1.
>
> **Policy Search.** Figures 9 and 10 refer to 40 episodes of 128 rollout samples each, resulting in 5120 total rollouts. Further details are in Table 6.
>
> **Figures 5 and 6.** $k$ is the number of elite samples in CEM. We have added this to the caption. $k$ also meant the desired ESS for ESSPS, but we have renamed this as $N^*$ to match the main text. We have also documented the meaning in the caption.
>
> We hope the revised draft has improved the remaining concerns. Please let us know if there are further issues.
>
> [1] Z. Wang, O. So, J. Gibson, B. Vlahov, M. S. Gandhi, G.-H. Liu, and E. A. Theodorou. Varia-
> tional inference MPC using Tsallis divergence. In Robotics: Science and Systems, 2021.

---

### Official Review · Reviewer_qNLg · 2022-07-30

**Originality:** Good
**Technical Quality:** Very Good
**Clarity Of Presentation:** Good
**Impact:** 3

**Recommendation:**

Weak Accept: I recommend accepting the paper, but will not argue for my recommendation if the majority of other reviewers have a different opinion.

**Summary:**

This paper considers Monte Carlo based optimal control through the lens of inference-based control whilst considering
the policy search, motion planning, and model predictive control (MPC) settings.
The paper has two main theoretical contributions.
Firstly, they introduce an adaptive scheme for dynamically setting the likelihood temperature parameter $\alpha$
in the Monte Carlo setting.
Their method is based upon optimising a lower bound on the expected return of the posterior based on the
importance sampled estimate using the prior.
Secondly, they propose a method for smoothing action sequence samples by considering a continuous-time Gaussian process
over the actions.
In contrast to the previous Monte Carlo approaches cited in the paper, their method produces smooth action trajectories
by capturing correlations between the time steps.
They test their methods in multiple simulated environments.


**Issues:**

1. Introduce the need for an adaptive temperature scheme more clearly.
    - Perhaps use a figure similar to the ones used on the website?

2. Figure 1 is never referenced. This figure is a nice motivation for smooth action sequences so use it!

2. Add more signposting or restructure Sections 2 and 3 to make them easier to follow.
    - There's a lot of maths here. Hold my hand and guide me through it...
    - I am a big believer in "telling the punch line first" when it comes to academic writing. I'd encourage the authors to try and follow this approach here.

3. When making statements about observations from figures (especially in the results section), walk me through what I should be looking at and make the statements so informative that I don't actually need to refer to the figure(s).

Minor fixes:

- Lemma 1: Does $\lambda$ represent Lagrange multipliers? This should be explicitly stated.
- Figure 1: Add x-axis ticks for reward. I had to use a ruler to compare iCEM and PPI.
- Line 4: missing "s" from "optimization"
- Line 12: remove "for"
- Line 53: "optimal setting" should be "optimal control setting"?
- Line 55: maximising the expected reward?
- Line 63: Consider replacing "concerns" with "considers"
- Line 63: "open-look" should be "open-loop"?
- Line 63: "f in an" should be "f is an"?
- Line 99: "This approach is an" should be "This approach is a"
- Line 129: "Reny-2" should be "Renyi-2"?
- Line 134: Add equation number for IS lower bound
- Line 154: Introduce $\mathcal{O}$ as a binary random variable $\mathcal{O} \in \{0, 1\}$ where $\mathcal{O}_{t}=1$ indicates time step $t$ being optimal and that you denote $\mathcal{O}_{1:t} = \{\mathcal{O}_i=1\}_{i=1}^{t}$. Readers unfamiliar with inference-based control may be confused otherwise.
- Equation below Line 154: time indices look wrong. If you want $q_{\alpha}(\alpha_{t:t+H} | ...)$ then shouldn't the integral be over $\text{d}\alpha_{1:t-1}$?
- Line 155: should $h$ be $H$?
- Line 208: "approximate applying" should be "approximate by applying"?
- Line 218: "popularized [18]" should be "popularized in [18]"?
- Line 219: Introduce acronym "PIPC"
- Line 240-242: Sentence doesn't read well. Should it say something like: "To understand the behaviour of the proposed posteriors as solvers, the performance results of LBPS and ESSPS on a range of standard black-box optimization functions over a range of hyperparameters, with e REPS and CEM as baselines, are shown in Appendix D.1."
- Line 242: "is an useful" should be "is a useful"
- Line 263: Introduce acronym "$\text{PI}^2$"
- Line 276: "lengthscale." should be "lengthscale fixed."?

- Line 11, 64, 223, 228, 285: No comma before "and"


**Quality Of The Limitations Section:**

Limitations are addressed clearly

**Reviewer Expertise:**

3: The reviewer is fairly confident that the evaluation is correct

**Robotics Focus:**

Highly relevant to robotics but no hardware experiments

**Strengths And Weaknesses:**


**Strengths**

1. The adaptive temperature scheme presented in this paper is well grounded theoretically whilst also performing well experiments.
    - Their method can be used for MPC, motion planning and policy search.
2. Their approach for modelling action sequences clearly results in smooth action sequences.
    - First time I have seen RFF features used to learn a policy.
3. Although I have less experience with Monte Carlo inference-based control,
    the paper appears to cite the relevant literature well.
4. I liked how the authors draw connections between methods that use the Gibbs posterior (Bayesian smoothing, maximum entropy, mirror descent and entropy-based RL).

**Weaknesses**


1. The motivation is not immediately clear.
    - The authors state that adaptive temperature schemes are crucial for optimization and that in PPI it has a strong influence on the posterior.
    However, they do not provide any references or intuition.
    - I think this forms a large motivation for the paper.
    - Interestingly, the motivation for the adaptive temperature scheme on their website is very intuitive and easy to follow.
    - I think Section 2 should contain a figure similar to the first figure from the website as it helps provide intuition about how the temperature $\alpha$ effects the posterior $q_{\alpha}$.

2. I found Sections 2 and 3 are hard to follow.
    - I think they need more signposting to keep the story consistent throughout Sections 2 and 3.
    - For example, Section 2 ends abruptly and I wasn't too sure about the main takeaways.
        - Why not add something like this to the start of section 3.

        > "In this section we introduce our method for adaptively setting the likelihood parameter. It builds on the connection between  INSERT CONCISE DESCRIPTION OF SECTION 3"

   - Further to this, I wasn't too sure why we were jumping from Theorem to Lemma in Section 3.
    I think the authors need to restructure this section or at least offer more signposting to help the reader through
    a section so heavy on mathematics.
    - I would be tempted to put Equation 5 at the start of section 3, then explain the meaning of the terms and
    show where they come from / how they are derived.

3. There are a few places where the authors offload work to the reader.
    For example, I often had to refer to two or three figures,
    often on different pages to understand a single statement made in the experiments section.
    Further to this, I often had to figure out what (sub)figures to check.
    Some examples:
    - Section 6.2 the authors write,

        > "The results in Appendix D.2 confirm that these solvers are all capable of solving the task, while RBF and RFF features perform equally well"

        - What exactly should I be looking at? The top rows of Figure 5:8 showing the expected return converging? Assuming this is what I'm meant to be looking at, why not write something like:

        > "Figures 5:8 in Appendix D.2 show the expected return converging for all of these solvers (for at least some parameter settings), which confirms they are all capable of solving the task. Also note that RBF and RFF features perform equally well."

    - Line 275: "Figure 13 replaces the SE kernel with..."
        - To understand this paragraph I had to read Figure 13 in the appendix to understand that the first two sentences were referring to MPC.
        - Why not write something like this:

        > "Figure 13 shows MPC results with finite feature approximation where the SE kernel has been replaced with RBF and RFF features."

    - Section 6 first paragraph: "Black box optimization compares..." - I had no idea this was referring to Section 6.1 until my second read of the paper. Similarly for "Policy search compares...". Consider changing these to "In Section 6.1 we compare" etc.

4. I found the equations for Lines 174:179 hard to follow. Why not tell the punch line first? Introduce the equation you want (equation 6) and then explain what the different terms are. In general, this writing style is much easier to follow.

5. The approach for obtaining smooth controls presented in this paper is elegant from a probabilistic
    modelling perspective, however, I am curious how it compares to encoding smoothness into the reward/cost function.
    For example, adding a quadratic cost term on the controls is commonly used in control to encode the notion of minimum
    effort/energy and is really simple to implement.
    I am not suggesting the method presented in this paper is not a useful approach, I am more suggesting that the authors
    should highlight how their method differs from (and hopefully improves on) adding an extra cost term.
    The authors touch on this in lines 289/290 but I am interested if they have any further thoughts?

**Summary Of Recommendation:**

Overall, I think this paper presents some nice work.
Their lower bound for optimizing the temperature seems to be grounded well from a theoretical perspective
and the results suggest that it works well in simulation.
Whilst I am less sure about the novelty of modelling the action sequence with continuous-time Gaussian processes,
their method does achieve smooth control and I liked seeing results using GP feature approximations.

Although the work seems strong, the paper suffers from two limitations.
Firstly, I do not think the need for an adaptive temperature scheme in PPI is introduced very well.
Interestingly, the motivation for the adaptive temperature scheme on their website is very intuitive and easy to follow.
The authors should consider putting some of this into the paper.

Secondly, I found the paper difficult to read.
This may be due to the amount of work the authors are trying to fit in and the quite obvious lack of space.
Nevertheless, I think this needs to be addressed.
In particular, I think Section 2 ends abruptly as I wasn't too sure what I was meant to take away from Section 2.
Further to this, the start of Section 3 does not detail what the section is going to be about.
As a result, I ended up being lost whilst jumping from Theorem to Lemma.
I think the authors need to restructure this section or at least offer more signposting to help the reader.

---

> ### Author Response · Authors · 2022-08-23
> **Rebuttal response to Reviewer qNLg**
>
> We wish to thank the reviewer for the extensive review and many helpful comments. We have been able to address the majority of comments.
>
> Regarding specific concerns
>
> **Clarity.** As discussed in the general response, we have moved some non-essential technical results to the appendix to make more room for passages that contextualize, summarize and connect each section (i.e. signposting). We have also rearranged some derivations when possible. The transition from Section 2 and 3 was revised to make the connections clearer.
> However, we have not restructured Section 3, as we find it easier to motivate LBPS starting from eREPS. We hope moving the lemmas to the Appendix improves the flow. We have also unified the notation across Section 2, 3 and 4.
>
> **Temperature motivation.** We included more motivation for the temperature in Section 3, by arguing for some issues episodic REPS. As requested, we have turned the animations from the project website into figures. Due to space constraints, they are included in Appendix A to support the quadratic Gaussian example in Example 1.
>
> **Smooth Prior vs Augmented Cost.** The reviewer is correct that there is a duality between our use of GP priors and augmenting the cost with action penalties. This is clear when viewing the Gibbs objective (Equation 3) where the KL between the prior and posterior is added to the task objective. For independent Gaussian priors with fixed covariances, this KL is equivalent to a squared penalty w.r.t. The mean (e.g. like weight decay in deep learning). Due to this aspect, prior work (e.g. MPPI) have combined the action penalty cost term and Gaussian action prior as their temperature is constant. In this work, we separate the action prior and any action penalty term in the likelihood, which simplifies the implementation. The key aspect of this work is the Gaussian process prior which is correlated in time, which therefore provides both amplitude regularization (due to Gaussian marginals) and smoothness regularization (due to the kernel / covariance function). This is easier to achieve at the solver-level, as we evaluate our methods on standard Gym MDP environments that have Markovian rewards, whereas a smoothness penalty in a reward would require the history of actions, e.g. $r(s_t, a_t, a_{t-1}, \dots, a_1)$. Secondly, while smoothness is easier for episodic rewards, it still requires turning the weight term between task objective and the additional action regularization. Our PPI perspective does this automatically using a temperature strategy. Thirdly, while we use the squared exponential kernel here, our Gaussian process approach is quite general. Smoothness can be controlled by both the lengthscale of the stationary kernel and kernel choice, for example Exponential, Rational Quadratic or Matérn kernels would provide different (e.g. rougher) smoothness regularization. Finally, by sampling from this Gaussian process, we are sampling action sequences that complement this penalty which also provide the Monte Carlo sample efficiency. This efficiency wouldn’t occur for the same smoothness penalty with independent action samples. We have added this discussion point to Appendix A.
>
> **Experimental results.** We have revised these sections to be more specific in referencing and describing results, as suggested.
>
> **Math.** We have rewritten the Gaussian process time-shift derivation in Section 4 as a proposition, so the final result is stated more clearly. For the LBPS objective, we have moved Lemma 1 and 2 to the appendix to improve the flow and make room for more supporting text, but we have kept the general structure the same as we wish to motivate LBPS starting from eREPS.
>
> We hope this revised submission addresses all concerns, please let us know if there are further issues

---

### Official Review · Reviewer_vNLR · 2022-07-31

**Originality:** Good
**Technical Quality:** Fair
**Clarity Of Presentation:** Poor
**Impact:** 2

**Recommendation:**

Weak Reject: I recommend rejecting the paper, but will not argue for my recommendation if the majority of other reviewers have a different opinion.

**Summary:**

The paper studies episodic posterior policy iteration method, specifically the questions of choosing the temperature hyperparameter of the Gibbs likelihood and inducing smoothness. The approach suggested by the authors exhibits better behavior in terms of smoothness and regret than the baselines considered.

**Issues:**

Clarity. See the “Strengths And Weaknesses” part of the review.

**Quality Of The Limitations Section:**

Limitations are addressed clearly

**Reviewer Expertise:**

1: The reviewer's evaluation is an educated guess

**Robotics Focus:**

Highly relevant to robotics but no hardware experiments

**Strengths And Weaknesses:**

**Disclaimer**. I may lack a sufficient background in control, my primary expertise is in Gaussian processes. These are a part of the paper present even in the title, but it is not the central part. I apologize for any poor judgment that may follow in my review, I keep my confidence score very low to reflect this.

Having that said, I spent an honest 7 hours trying to comprehend the paper, without much success. The paper builds on a huge amount of prior work (there are almost a hundred references!) and does not seem to introduce the necessary background well. To be honest, on several occasions it seemed to me as if the paper was specifically composed by putting as much of the specialized terminology/methods or claims from the literature on an inch of paper as it is possible.

Let me be more specific. First, at the end of page 2, the Gibbs posterior is introduced. It is immediately cast to an optimization problem. Then, on page 3, this optimization problem is replaced by another, now a constrained optimization. So far so good, I lose track after, at: “we reverse the objective and constraint and adopt an equality constraint”. Why do we do this? Then the argument proceeds by approximating some constraints (several times), replacing something by its lower bound in the concentration inequality, all this using multiple results from the literature. And in the end it is absolutely not clear what we have obtained. Are we now able to automatically tweak the temperature hyperparameter? Did we prove/infer something or was the argument just a heuristic motivation? In the former case, it obviously lacks rigor. In the latter case, it lacks motivation and explanations for each of the steps. If these would have taken up too much space, maybe it is worth extending the paper to the full-sized journal manuscript? Otherwise, maybe formulating some of the results as theorems may help, or explicitly stating the intermediate results and their meaning in the context of the whole paper.

On the other hand, I would like to mention that a software implementation is in the supplementary, which I think is a huge plus.

**Small Comments**
1. Line 12. “for to” -> “to”.
2. Figure 1. There is no legend, it is not clear e.g. what a_0 is. Furthermore, this figure is first referred to from the main text on page 8. For a reader who only just read page 1, where the figure is located, it is not clear what the abbreviations mean.
3. Equation (1). Is there a reason to write “a_1, a_2, … ” instead of “a_1, a_2, …, a_T”? The former notation is usually reserved for infinite sequences.
4. Line 56. Maybe “setting, performs” -> “setting, i.e. performs”?
5. Line 57. In “R(S, A)”, the “S” was not formally defined.
6. Line 61. “Q” is not defined.
7. Line 63. Probably “in an episodic return” -> “is an episodic return”.
8. Line 121: “is an hyperparameter” -> “is a hyperparameter”.
9. Equation in Lemma 2. \mathbb{D}_2 seems to be undefined.
10. Line 144. Maybe “posterior to one” -> “posterior to the one”?
11. Line 242. Maybe “is an useful” -> “is a useful”?
12. Figures 3 and 4. It would be good to explain what smoothness means in the caption.

**Summary Of Recommendation:**

The paper was extremely hard to read for me and after spending 7 hours trying to understand it, I failed. It may very well be caused by a misalignment between my expertise and the subject of the paper, hence my low confidence. However, I cannot vote for acceptance of a paper I am sure I have not understood.

---

> ### Author Response · Authors · 2022-08-23
> **Rebuttal response to Reviewer vNLR**
>
> We wish to thank the reviewer for their honest and extensive feedback, as well as the time invested in this review. We agree that the initial draft had numerous clarity issues, especially for those unfamiliar with the prior literature, and we have uploaded a new draft that we hope provides much greater clarity.
>
> To discuss specific points:
>
> **Clarity.** Please refer to the main comment. Specifically, we have substantially revised Section 2 and 3 to provide more of a context and motivation for the mathematical steps highlighted in your review. We also hope that by moving Lemmas 1 and 2 to the appendix, this section is less overwhelming to the reader.
>
> **Temperature motivation.** As we show in Definition 1, Appendix A and the project webpage, the inverse temperature $\alpha$ defines a family of possible posterior distributions $q_\alpha$. The question is which value should be used for Monte Carlo optimization? Some prior work use a constant (e.g. MPPI) which requires extensive tuning and offers no inuition. Other works adopt heuristics that reduce the need for tuning, but are still not theoretically motivated. Table 1 in the Appendix summarizes these strategies. We argue REPS is a principled approach for optimization, as the temperature is defined by a hard KL constraint which can be interpreted as a trust-region from the constrained optimization literature. However, this hard KL bound is hard to specify, and in the Monte Carlo setting it is satisfied through a Monte Carlo approximation. This means that in the low sample setting, you are tuning for KL bound that is very approximately satisfied. You can see this in Figure 9, where the KL constraint does not match the actual KL divergence between updates. Note that REPS approaches do not have this issue in the parametric setting, for example MORE [1] and Guided Policy Search [2], where the KL term can be computed exactly. To resolve this issue for the Monte Carlo setting, we were inspired by the cross entropy method (CEM), which is parameterized by the number of elite samples. This is an effective approach that is comparably much easier to specify and more intuitive, which we believe is why CEM is so popular as a stochastic optimization method. To connect CEM to REPS, we investigated the connection via the effective sample size (ESS), which led us to the Renyi-2 divergence result. While the connection between the ESS and Renyi-2 divergence turned out to be a known result, we believe it is still not widely known and we think it provides an interesting connection between PPI / REPS and CEM approaches. While we do not offer convergence guarantees, we believe this is a principled derivation to a solution that answers the temperature question we posed above. To improve this clarity, we have worked these discussion points throughout the new paper draft.
>
> **Figure 1.** Thanks for raising this. This figure is designed to be a pull figure to motivate all the theory to the eventual complex robotic simulation experiments. We have improved this figure to be clearer in this regard, to show that we use our method to capture aspects of, and improve upon, baseline algorithms. We also reference this figure in the introduction to work it into the main text.
>
> **Smoothness definition.** The definition of smoothness is somewhat involved and takes three sentences to define (277-280). Therefore, we believe it is too lengthy to be included in all relevant figure captions due to space constraints. As a compromise, we have added a hyperlink from the caption to the definition, to aid those reading digitally. We have also added arrows to the labels to indicate which values are better. We also want to highlight that we include the definition at the start of the MPC experiment section, which is the most intuitive place for readers to find it.
>
> We hope that revised text has improved clarity and motivation sufficiently for more unfamilar readers such as yourself. Please let us know if there are further issues.
>
> [1] A. Abdolmaleki, R. Lioutikov, J. R. Peters, N. Lau, L. Pualo Reis, and G. Neumann. Model-  based relative entropy stochastic search. In Advances in Neural Information Processing Systems, 2015
>
> [2] Learning complex neural network policies with trajectory optimization
> S Levine, V Koltun - International Conference on Machine Learning, 2014

---

### Official Review · Reviewer_VNXb · 2022-08-07

**Originality:** Good
**Technical Quality:** Excellent
**Clarity Of Presentation:** Fair
**Impact:** 4

**Recommendation:**

Strong Accept: I recommend accepting the paper and will argue for my recommendation even if other reviewers hold a different opinion.

**Summary:**

This paper proposed a novel method for smooth control in Model Predictive Control (MPC). It achieves better smoothness by incorporating Gaussian Process action priors in inference-based control. In order to reduce computation in GP, it factorized the covariance matrix by matrix Normal distribution [58]. Also, it proposed two new algorithms (LBPS and ESSPS) for optimizing the temperature in the Gibbs posterior, based on regularized Importance Sampling (IS) approximations. The effectiveness of the proposed algorithms as well as the smoothness are verified in simulation benchmark tasks.


**Issues:**

Please make the setting regarding the use of GT model clear.

Please see the weaknesses section for other issues.

Question:
Perhaps i misunderstood it, but I thought that the GPMP [30] only used GP to model state sequence. But the paper suggested (below line 158) that it was used to model the policy. Is that right?


**Quality Of The Limitations Section:**

Additional details required

**Reviewer Expertise:**

2: The reviewer is willing to defend the evaluation, but it is quite likely that the reviewer did not understand central parts of the paper

**Robotics Focus:**

Relevant but unlikely to deploy to hardware in near future

**Strengths And Weaknesses:**

### Strengths

- Novel use of GP for action prior with efficient implementations for high dimensional control
- Abundant ablation studies
- The proposed methods were able to achieve a better smoothness empirically
- Leveraged the IS lower bound [48] to form a new objective for temperature adaptation. The proposed perspective of inference-based control based on Gibbs posterior unified some previous methods

### Weaknesses

**Experiments**

The experiments are based on having access to the ground truth model but this setting is only vaguely mentioned in the last two lines of the paper… “vague” as in “future work will consider… additional consideration of dynamics approximation”. Please be explicit about the setting.

LIne 271-272: SE leads to better smoothness “by a factor of 2 compared to baselines”, but it doesn’t seem to be the case for MPPI in Fig.4. Am i misinterpreting it?

Line 275 discusses the results from Fig.13. But the figure does not show SE Kernel results despite having it in the legend.

**Presentation**

The paper appears a bit unorganized. It covers a lot of technical background as well as the technical contribution but I feel that the contribution is a bit scattered throughout the text. As a result, the contribution of the paper is a bit obscured and takes a lot of time for the reader to parse.

**Methods**

- I missed it but I did not find the descriptions on how to choose N* in ESSPS.
- Also, I could not fnid how the SE kernel parameters were chosen.

**Minor**

- Fig. 5,6,7,8: what do the parameters mean?

- The adaptive temperature strategy in Sec. 3 appears to be a bit disconnected from the policy iteration algorihtm, without reading further into Appendix A and B. It’s partly from the switch of notations at the beginning of Sec 4 (using O to represent the Gibbs likelihood). It would be better to explicitly write it out.

- Line 685: should -> showed?



**Summary Of Recommendation:**

The paper seems to contain decent contribution: inference-based control perspective based on Gibbs posteriors and new methods for temperature adaptation and smooth control.
I think that the presentation could be improved though, as explained in the weakness section.

=======
post-rebuttal
=======
I've raised the score to strong accept to reflect what I mentioned earlier in the rebuttal response.

---

> ### Author Response · Authors · 2022-08-23
> **Rebuttal response**
>
> We wish to thank the reviewer for their helpful comments. We were able to incorporate all the suggestions into the draft.
>
> Regarding specific points:
>
> **Clarity.** We discuss this in the general response. Specifically, we have improved the connection between sections, connecting the temperature tuning to GP priors, and fixed the math notation in Section 4 to be more consistent with Section 3.
>
> **Hyperparameters.** We discuss hyperparameter selection in Section E of the Appendix. In summary: we reused hyperparameters from prior works when relevant, otherwise we performed grid search. For ESSPS, we chose N* to match the number of elite samples used by the CEM baselines. This essentially made ESSPS an ablation between hard (i.e CEM) and soft (i.e. PPI) weights for a given ESS.
>
> **Smoothness.** Re: Figure 4, we have clarified the statement to ‘up to a factor of 2’, as the smoothness improvement varies with samples.
>
> **Oracle dynamics in MPC.** Thanks for highlighting this. We have added the ground truth aspect to the dynamics to the MPC subsection title to avoid confusion. Moreover, we do not mention model-based reinforcement learning (MBRL) in the main text at all, apart from referencing MBRL papers. As the policy search experiments are performing reinforcement learning, the use of ground-truth dynamics is not a general assumption of the paper, only for the MPC section.
>
> **GPMP.** The reviewer is correct that GPMP is used for motion planning, where the GP defines a state trajectory. However, in many policy search settings (e.g. our ball in a cup experiment), the policy parameterized a desired state trajectory that a lower-level controller tracks, and ProMP policies can also be conditioning on desired initial and terminal states. Therefore, the two settings are not so different in practice. However, we have removed this line since it is a source of confusion.
>
> **Appendix Figures.** Thanks for flagging this, we have expanded these figure captions to make them more informative.
>
> We hope this new revision fixes the organization and clarity issues. Please let us know if you have further concerns.

---

> > ### Comment · Reviewer_VNXb · 2022-08-27
> > **Re: Rebuttal**
> >
> > I thank the authors for the clarifications and I appreciate the effort during the rebuttal to improve the paper.
> > My concerns have been adequately addressed. The clarity of the revised paper (main and appendix) has improved quite a bit IMHO.
> > I don't seem to have access to edit the score at the moment but I'm increasing my score to a strong accept. Great work!

---

### Meta-Review · Area_Chair_AtcL · 2022-08-12

**Recommendation:** Accept (Oral)
**Confidence:** 5

**Metareview:**

The paper received overall positive reviews with reviewers commenting on the well developed theory proposed and the extensive set of experiments, including ablation studies. The main issues to be addressed by the authors are:

1. Organisation and clarity. The paper can be made significantly clearer, in particular the specific contributions
2. Clearly separate control and inference to make the paper more accessible to readers with different backgrounds
3. Address the multiple comments on the motivation for smooth trajectory priors
4. Clarify the need for the adaptive temperature scheme

=================================

Post rebuttal update

The authors have significantly improved the clarity of the paper which is now accessible by a broader audience. They have also clarified the motivation for smoothness and the adaptive temperature scheme. The paper contains a solid contribution and I recommend acceptance.

**Best Paper Nomination:**

No